# Learning Dense Features for Point Cloud Registration Using a Graph Attention Network

**Quoc-Vinh Lai-Dang \***, **Sarvar Hussain Nengroo** and **Hojun Jin**

Cho Chun Shik Graduate School of Mobility, Korea Advanced Institute of Science and Technology (KAIST), 291 Daehak-ro, Yuseong-gu, Daejeon 34141, Korea; sarvar@kaist.ac.kr (S.H.N.); hjjin1995@kaist.ac.kr (H.J.)
**\*** Correspondence: ldqvinh@kaist.ac.kr; Tel.: +82-42-350-1287

**Abstract:** Point cloud registration is a fundamental task in many applications such as localization, mapping, tracking, and reconstruction. Successful registration relies on extracting robust and discriminative geometric features. Though existing learning-based methods require high computing capacity for processing a large number of raw points at the same time, computational capacity limitation is not an issue thanks to powerful parallel computing process using GPU. In this paper, we introduce a framework that efficiently and economically extracts dense features using a graph attention network for point cloud matching and registration (DFGAT). The detector of the DFGAT is responsible for finding highly reliable key points in large raw data sets. The descriptor of the DFGAT takes these keypoints combined with their neighbors to extract invariant density features in preparation for the matching. The graph attention network (GAT) uses the attention mechanism that enriches the relationships between point clouds. Finally, we consider this as an optimal transport problem and use the Sinkhorn algorithm to find positive and negative matches. We perform thorough tests on the KITTI dataset and evaluate the effectiveness of this approach. The results show that this method with the efficiently compact keypoint selection and description can achieve the best performance matching metrics and reach the highest success ratio of 99.88% registration in comparison with other state-of-the-art approaches.

**Keywords:** deep learning; graph attention network; point cloud registration; Sinkhorn algorithm





## 1. Introduction

Efficient keypoint matching is required for different applications such as 3D object identification, simultaneous localization and mapping (SLAM), sparse depth odometry, 3D shape retrieval, and range image/point cloud registration [1–4]. All of these techniques include keypoint recognition and their matching by employing element descriptors to find genuine keypoint correspondences. To find salient/interesting points on 3D point clouds, a variety of 3D keypoint detectors are available in the literature [5,6]. Following the detection of keypoints on 3D point clouds, feature descriptors are used to match them. Feature descriptors are multi-dimensional vectors that encode information in the vicinity of a keypoint. In most cases, the feature descriptors are matched by computing Euclidean distances in their high-dimensional vector spaces, which is a computationally expensive procedure. Furthermore, the memory footprint and computing requirements for feature descriptor matching grow with the size of the feature descriptor.

Different handheld depth sensors such as Microsoft Kinect [7], Asus Xtion Pro Live [8], Intel RealSense camera [9], and Google Tango [10] have made it possible to access 3D data. Since then, there has been an increase in consumer applications that use 3D sensors and interpret dense depth data on mobile devices for vision and robotics. Project Tango, a Google ATAP project that can give online 3D posture estimates and detailed depth data from a mobile device, has been in the spotlight. The accessibility of these transportable 3D data acquisition devices necessitates the development of applications with a low memory footprint and low computing power requirements.

Wireless sensor networks (WSNs) are being applied not only in the traditional way with a limited number, but also allow connecting a large number of high-quality sensor network nodes for practical problems as shown in Figure 1. Technical issues of the WSNs are broad, including spectrum sensing [11], broadband data transmission [12], and recharging rechargeable WSNs [13,14]. Due to their rapid development in recent years, deep learning techniques allow us to improve the performance of sensor networks used in communication protocols, intelligent transportation systems, and smart charging schemes, especially for vehicles [15,16]. To date, light detection and ranging (LiDAR) is considered one of the most outstanding sensor systems helping autonomous vehicles with perception of their surroundings. LiDAR scans the objects creating sets of data called point clouds, which are continuously aligned with each other to create 3D models through point cloud registration.

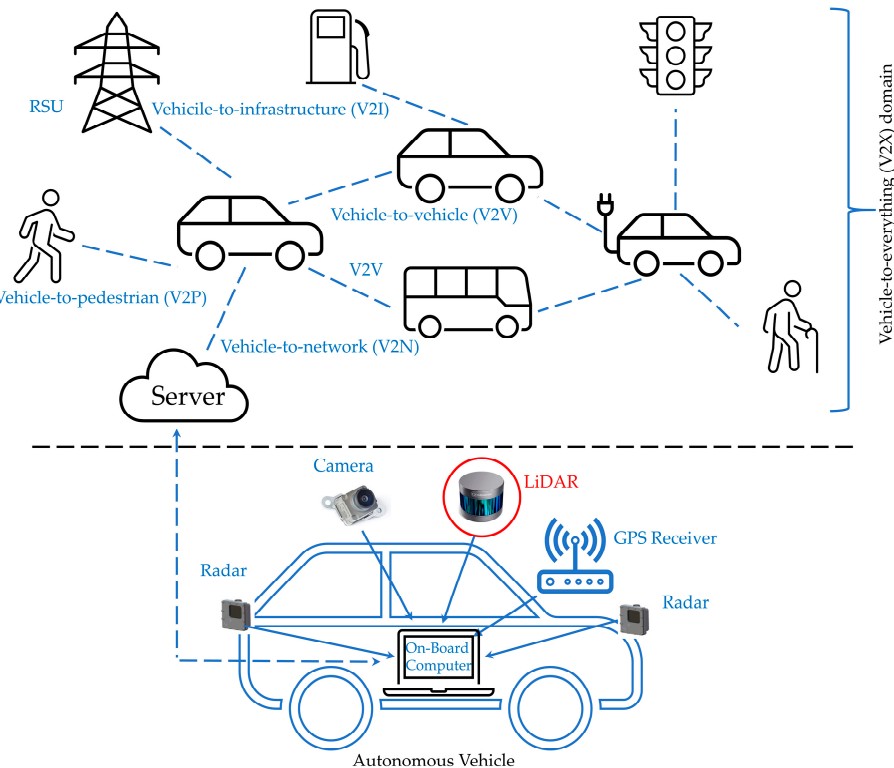

**Figure 1.** Autonomous vehicles and their wireless sensor networks.

3D point cloud registration plays a fundamental role in robotics applications. Many systems use it for applications of autonomous driving, such as LiDAR-based SLAM [17,18], or archaeological inspection via 3D model reconstruction. Given two partially overlapped sets of 3D points, the goal of registration is finding the best Euclidean relative transformation to align these fragments according to a common frame, thus obtaining a successful point cloud registration. Existing solutions can be categorized into two main methods. Methods that iteratively minimize the distance between nearest-neighboring points with good initialization, e.g., iterative closest point (ICP) [19] and its variants [20–22] are popular approaches. Other methods randomly sample corresponding points through robust descriptor matching, e.g., using random sample consensus (RANSAC) [23]. Unfortunately, the key challenge of finding 3D-3D data correspondences is non-trivial.

This problem is particularly challenging because (1) different point clouds can be captured with different angles, and (2) the raw 3D scans are noisy and have various densities; thus, descriptor-based matching often increases robustness across different scenarios. To compute compact 3D descriptors, there are one-stage [24–28] or two-stage [29–31] methods. The former directly encodes the local geometric information of raw points (e.g., collection of coordinates). On the other hand, the latter firstly determines a local reference frame

(LRF) from the points to transform the whole patch into suitable representations; they then encode the information of these new representations into a generalized descriptor.

Recent descriptors based on deep learning [25,26,32] have outperformed their hand-crafted counterparts [29,33] because of the availability of large-scale labeled datasets. The methods based on one-stage learning can achieve rotation invariance by encoding the local geometric information with a set of geometric relationships, and then by learning descriptors via a deep network to achieve generalization ability and permutation invariance. In two-step methods, the LRFs can be computed and transformed into a voxel grid, where each voxel contains the information of inside points. Finally, descriptors can be encoded from the voxelization by using ConvNet [34].

The SpinNet [35] is a new neural architecture which projects the point cloud into a cylindrical space; then the descriptor is invariably extracted rotationally by a special convolutional neural layer. Inspired by the success of GAT for computer vision tasks, Shi et al. [4] proposed a multiplex dynamic graph attention network matching method (MDGAT-matcher). It uses a novel and flexible graph network architecture on the point cloud and achieves the enriched feature representation by recovering local information. Although MDGAT-matcher finds high-quality data associations, its architecture provides space for improvements. Without employing the rotation-invariant description method, it limits the robustness and distinctiveness of the learned descriptor. In this study, we aim to tackle these challenges in a joint learning framework that is able to not only accurately detect keypoints from 3D point clouds, but also derive compact and representative descriptors for matching.

To this end, we find inspiration in the MDGAT-matcher [4] for a GAT-based two-stage descriptor for point cloud registration. However, the extension of our work is non-trivial. A reliable network for the extraction of robustness features against noise and occlusion is needed instead of using hand-crafted counterparts. In this work, we adopt joint learning of dense detection and description of 3d local (D3Feat) [28] to extract dense features from the kernel point convolution (KPConv) network [36] and achieve density invariance on irregular 3D point clouds. Based on these density-invariant saliency scores, we detect keypoints in their local neighborhood. Finally, these dense features are further fed into the attention network, DFGAT, which fully exploits the local information and generates a generalizable and distinctive feature vector.

Overall, the proposed DFGAT has the following key features:

- It is the learning pipeline using joint keypoints descriptors and detectors, is density invariant, and generalized across all point clouds.
- It can build the global spatial relationship and enrich characteristics for matching features.
- It helps to save time and computing resources but still achieves the highest matching scales and success ratio in the registration experiment.

## 2. Related Work

### 2.1. Keypoint Detector

Hand-crafted 3D keypoint detectors are still commonly used even as the use of deep learning to generate keypoint descriptors is becoming more frequent [32,37]. A comprehensive review and evaluation of such common approaches can be found in [38]. The work in [33] detects outstanding keypoints with a large variation in the principal direction. Local surface patches [39] and shape indexes [40] define a keypoint based on the global extremum of its principal curvature in a neighborhood. Laplace–Beltrami Scale-space (LBSS) [41] computes the saliency by applying a Laplace–Beltrami operator on increasing supports for each point. Typically, the local spatial characteristics greatly influence the design of traditional methods applied to point clouds. Hence, the performances when these detectors are applied to the real world in 3D domains may be impaired by influences such as noise, varying densities, and occlusion. To improve detection ability, a recent study used an unsupervised stable interest point (USIP) [42], which introduces an unsupervised method to learn points that are likely to repeat and localize correctly. Additionally, this detector

is more robust because of its learning ability from data disturbances. Nonetheless, its performance is significantly degraded with a limited number of keypoints. This motivates the use of joint detection and a description pipeline to identify stable 3D local keypoints for matching.

### 2.2. Local Descriptors

Handcrafted descriptors consist of two groups: one-stage methods and two-stage methods. Typically, one-stage (Non-LRF-based) descriptors compute descriptors from point representations that are defined between pairs of points using 3D point coordinates only [43] and/or surface normals [29]. Point-pair features (PPF) [44] and fast point feature histograms (FPFH) [24] build an oriented histogram by using pairwise combinations of surface normals. They provide accurate feature matching with reasonable computational efficiency. The lack of essential geometrical information for the local frame is the main disadvantage of these methods. The two-stage (based on the LRF) descriptors, such as intrinsic shape signatures (ISS) descriptor [33], signature of histograms of orientations (SHOT) descriptor [30] and rotational projection statistics descriptor [45], are not only able to characterize the geometric patterns of the local support region but also effectively exploit the 3D spatial attributes. However, the LRF-based methods inherently introduce rotation errors, sacrificing the feature robustness.

### 2.3. Graph Attention Network

Recently, the development of the GAT [46] contributes to simplifying CNN into graph representations. Even though there are often common traits with on-set deep learning, the GAT can aggregate more powerful features along the edges and nodes. Previously, Zanfir et al. [47] presented an application of deep neural networks for finding the similarity matrix in graph matching. Wang et al. [48] introduced another GAT-based network for the image domain. Its framework consists of a feature extractor in the image, a graph embedding component, an affinity metric function, and a permutation prediction. Typically, the graph matching problem is mathematically formulated as a quadratic assignment problem, which consists of finding the assignment that maximizes an objective function of local compatibilities (a linear term) and structural compatibilities (a quadratic term) [47,49]. Many algorithms have been proposed to find a locally optimal solution for the above matching problem. One popular method is to use an approximation of the objective function. By using the Hungarian method proposed by Kuhn [50], the problem can be solved for minimizing total cost (or) maximizing the profit. As an approximate version of the Hungarian method, a Sinkhorn network [51] is developed for linear assignment learning with a predefined assignment cost which is designated to enforce doubly-stochastic regulation on any non-negative square matrix. However, very few studies have focused on using GAT to solve the point cloud registration problem. The proposed methodology using GAT improves performance but still saves the computational cost of the registration point cloud.

In the MDGAT-matcher [4], a dynamic GAT was introduced to embed the geometry feature from the FPFH descriptor to create more robust feature for the matching process. Because of the limitations of handcrafted descriptors, their results are sensitive and rotation-variant to rigid transformation in Euclidian space. Recently, Xuyang et al. [28] proposed D3Feat which is invariant in density, representative, and has outstanding adaptive capability across untrained scenarios. However, it either requires the calculation of the point density or relies on random points selection to achieve rotation robustness. This is often unfeasible and is insufficient when there are limited computing resources. However, the proposed method not only selectively exploits the rich spatial and contextual information of the D3Feat, but also takes advantage of the flexible GAT to find high-quality point-wise correspondences in 3D point cloud feature matching.

The main focus of this study is to use a learning-based approach to overcome the limitations of manual methods. For indoor datasets, early works such as 3dMatch [25]

use a 3D convolutional network to learn geometric features by training on positive and negative pairs. The fully convolutional geometric features approach (FCGF) [27] used a fully convolutional network for descriptors and achieved rotation invariance. The SpinNet approach [35] used a spatial point transformer to project the input point to a cylindrical space followed by a series of powerful neural networks. It learns rotation-invariant, compact, and highly descriptive local representations. Similar to the FCGF, the D3Feat approach uses a fully convolutional network [28], but unlike FCGC, the D3Feat extracts global descriptors using dense deep features through a KPConv backbone [36]. Yew and Lee [52] used a weakly-supervised network to extract features in outdoor scenes. However, these descriptors are the result of random or full sampling. To overcome the above drawbacks, an effective combination of a keypoint and a descriptor is proposed to provide better registration performance.

## 3. Proposed Methodology

The overview of our DFGAT model is illustrated in Figure 2 and it consists of three main parts: the keypoint detector in Section 3.1, the descriptor encoder in Section 3.2, and the dynamic GAT in Section 3.3.

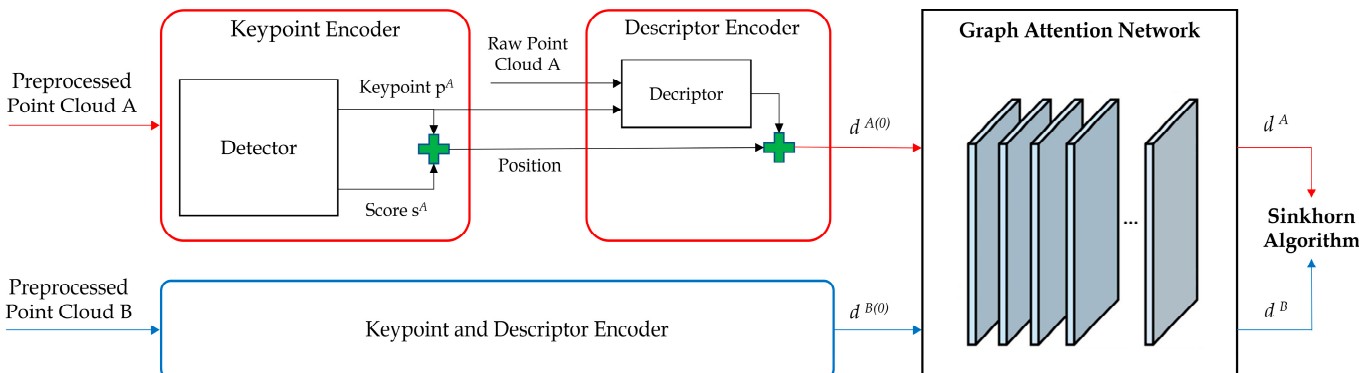

**Figure 2.** DFGAT pipeline. The architecture is made up of three major components: keypoint encoder (Section 3.1), descriptor encoder (Section 3.2), GAT (Section 3.3).

### 3.1. Keypoint Encoder

Instead of randomly choosing the local points as the input, we use a detector inspired by the D3Feat detector [28] to find the keypoints with local-max scores in the neighborhood within a radius. We denote a query point cloud from the training dataset as $Q \in \mathbb{R}^3$ and the dense feature matrix $F \in \mathbb{R}^d$, where $d$ is the dimension of feature. The condition of a point $p_i^Q$ to be a keypoint of point cloud $Q$ is given by:

$$\begin{cases} k = \underset{t}{\mathrm{argmax}} F_{i:k}^t \\ i = \underset{j \in N_{p_i^Q}}{\mathrm{arg\,max}} F_{j:k}^k \end{cases} \tag{1}$$

where $F_{i:k}$ is the $k$-th channels ($k = 1, \dots, d$) of the $i$-th descriptor corresponding with point $p_i^Q$ detected by the D3Feat detector, and $N_{p_i^Q}$ is the set of radius neighborhood points of $p_i^Q$. The maximum channel is chosen before comparing with the same channel of local neighborhood points. Xuyang et al. [28] used two local scores: Channel max score and Density-invariant saliency score. More details about these scores can be found in their work [28]. Both of these are used to build the final score map for each input of the 3D point cloud. We select the desired number of keypoints with the highest scores $s_i^Q$. Inspired by the SuperGlue [53], we encode each D3Feat keypoint position with its detection score into a higher dimension feature by using a multi-layer perceptron (MLP). The keypoint encoder

provides a better signature position and keeps spatial information around a keypoint in a form as follows:

$$pos_i^Q = MLP_{pos}\left(D3Feat\left(p_i^Q, s_i^Q\right)\right) \tag{2}$$

where $pos_i^Q$ is the encoded information about position and score of keypoint $p_i^Q$.

### 3.2. Descriptor Encoder

Inspired by the KPConv [36] in 2D convolution, Xuyang et al. [28] performed feature extraction on irregular point clouds. They added a new normalization term to ensure that the convolution formulation is robust against spatial density. Given the set of features $F_{i:n}$, the convolution by kernel $g$ at point $p_i^Q$ is given by:

$$(F_{i:n} * g) = \frac{1}{\left|N_{p_i^Q}\right|} \sum_{p_j \in N_{p_i^Q}} g(p_j - p_i^Q)f_j \tag{3}$$

where $p_j$ is a supporting point, and $f_j$ is the corresponding feature.

Inspired by the MDGAT-matcher [53], we combine the D3Feat of the $i$-th feature $F_{i:}$ with the output of the keypoint encoder to fully exploit spatial clues, and turn them into a new descriptor $d_i^Q$ as follows:

$$d_i^Q = MLP_{desc}\left(D3Feat(F_{i:}^Q, pos_i^Q)\right) \tag{4}$$

### 3.3. Dynamic Graph Attention Network

After the feature encoder, we apply a dynamic graph attention network, inspired by the GAT [54], to exploit relationship information for matching. In contrast to the traditional static architecture, the MDGAT-matcher [4] showed that a dynamic graph can deal with variable sizes of inputs and focus on the parts that matter most to make decisions efficiently. Furthermore, this technique operates a specific aggregator over a fixed-size neighborhood and yields impressive performance across a large-scale dataset.

#### 3.3.1. Graph Attention Layer

The GAT starts with a high-dimensional state for each node and computes at each layer an updated representation by simultaneously aggregating messages across all given attentions for all nodes. We use a single layer throughout the architecture and the following is a specific description of this graph attention layer. The input to our layer is a set of embeddings from the previous step which are used as node features, where the number of nodes is the number of keypoints, and the number of features in each node is the dimension of the descriptor vector according to Equation (4). To transform the input features into higher-level features, at least one learnable linear transformation is required. Moreover, our graph includes two types of attention: self-attentions and cross-attentions. Self-attention is a connection mechanism between keypoints in the same point cloud that utilizes surrounding information to increase their representation. On the other hand, cross-attentions connect keypoints between two-point clouds to inspect the valid connection of respective points. The layer produces a new set of node features as its output.

#### 3.3.2. Dynamic Graph Update

We inject the graph structure into the mechanism by performing masked attention. Within the same layer, for a query node $d_i^Q$ in query point cloud $Q$ and all neighborhoods source nodes $d_j^S$ in source point cloud $S$, the network computes three attention coefficients: query $q_i$, key $k_j$, and value $v_j$, which are used for later graph updating and attentional

propagation. To that end, a shared linear transformation, parametrized by weight matrices $W_{1,2,3}$ and biases $b_{1,2,3}$, is applied to every query node, as follows:

$$q_i = W_1 d_i^Q + b_1$$
$$\begin{bmatrix} k_j \\ v_j \end{bmatrix} = \begin{bmatrix} W_2 \\ W_3 \end{bmatrix} d_j^S + \begin{bmatrix} b_2 \\ b_3 \end{bmatrix} \tag{5}$$

We normalize the coefficients between different nodes by using the softmax function for easy comparison, i.e., $\alpha_{ij} = \text{softmax}_j(q_i^\top k_j)$. This weight shows how important the node $d_j^S$ is to the node $d_i^Q$.

To stabilize the learning process, we extend our mechanism to employ multi-head attention, similarly to Vaswani et al. [55]. Specifically, *K* independent attention mechanisms execute the transformation, and then their features are concatenated, resulting in the output feature representation. To avoid the bad influence of many dissimilar neighbor source nodes, we only extract certain *K* edges for each query node thanks to dynamic graph attention weight $\alpha_{ij}$. A decay strategy is adopted to reduce the links when the layers pay more attention to a specific area of interest.

### 3.3.3. Attentional Propagation

Once the graph updating is completed, attention propagation is performed on each node in the new graph via the message passing process as follows:

$$d_i^Q = d_i^Q + \text{message} \tag{6}$$

When the aggregation is completed to the final layer, the current descriptors are passed through an additional linear layer with weight matrix $W_4$ and bias $b_4$:

$$d_i^Q = W_4 d_i^Q + b_4 \tag{7}$$

### *3.4. Gap Loss*

To optimize feature embeddings, different loss formulas, such as contrastive [27] and triplet [32] are used. Similar works relevant to data metric learning consider effective negative exploitation [27]. The model is trained to pull similar features together while pushing dissimilar features apart. Considering a set of triplets of input histograms $p_i^Q$, it includes an anchor point in *Q*, a true connection in *S* that is known to be similar to the anchor, and a false connection in *S* that is known to be dissimilar, indexed by *i*, *p*, *n*, respectively. We select anchor points and a mining group per scene. A positive matching *p* is the matching with minimum Euclidean distance and shorter than the threshold value, whereas the negative matching *n* is any of non-positive matching for the keypoint in the anchor point cloud. In addition, we use the pairwise loss for the mined quadruplet $(p_i, n_i, p_j, n_j)$ and form a gap loss. Given the number of keypoints in these two clouds *M* and *N*, the ground-truth correspondences matrix $M \in \{0,1\}^{(M+1) \times (N+1)}$ are generated by projecting the keypoints to the other point cloud. Specifically, for each keypoint in the point cloud, it is either connected with a matched keypoint or with the non-matched keypoint (the so-called "dustbin") of the other point cloud. Consequently, the matching scores between any keypoint and the dustbin are filled in the last row and column of correspondences matrix *M*. For a generalized graph matching, we create the predicted soft score assignment matrix $\overline{P} \in [0,1]^{(M+1) \times (N+1)}$ using the Sinkhorn algorithm [53]. The loss is then calculated as:

$$Loss = \sum_{i=1}^{M} \log \left( \sum_{n=1}^{N+1} \left[ (z_i)_+ + 1 \right] \right) + \sum_{j=1}^{N} \log \left( \sum_{m=1}^{M+1} \left[ (z_j)_+ + 1 \right] \right) \tag{8}$$

where $z_i = -\log r_i + \log n_i + \eta$, $z_j = -\log c_j + \log n_j + \eta$, and $(z_i)_+ = \max(z_i, 0)$. Specifically, $r_i = \sum\limits_{(i,n) \in \mathcal{P}} \overline{P}_{in} M_{in}$ represents the true connection assignment value for keypoint $p_i$ in point cloud $Q$ and $\mathcal{P}$ is a set of all positive pairs in extracted features in a minibatch. Meanwhile, $n_i = \sum\limits_{(i,n) \in \mathcal{N}} \overline{P}_{in} M_{in}$ refers to the false connection assignment value for $p_i$ and $\mathcal{N}$ is a negative subset of features in a minibatch that will be used for negative mining. The gap loss aims to expand the margin between the positive match with all the other negative matches at least a margin away $\eta$. The second terms in the loss function for keypoint $p_j$ in point cloud $S$ are computed in a similar manner.

## 4. Implemented Details

### 4.1. Dataset

We perform several evaluations of our framework with the KITTI odometry dataset [56]. This is a large point cloud dataset obtained by the Velodyne-64 device while driving in diverse situations. This dataset includes 11 sequences of real driving situations. Inspired by the preprocessing data method in the MDGAT [4], we use sequences 0 to 9 for the training set, except sequence 8 for the validation set, and 10 for the test set. We follow D3Feat method [28] to utilize only scan couples with more than 30% similarity.

### 4.2. Parameter

We use the D3Feat model [28] pre-trained with outdoor LiDAR [56] and the dataset prepared by 3DFeat-Net [52] for keypoints selection and description. Specifically, we use 23,190 point clouds in the KITTI dataset that have been preprocessed to 256 desired keypoints/point cloud to evaluate the model. For each selective keypoint, we extract its descriptor by using the D3Feat pre-trained model. The output size of the keypoint encoder and descriptor encoder are 128. For the architecture of the GAT, we use $L = 9$ layers of alternating multi-head self-attention and cross-attention with four attention heads and perform $T = 20$ Sinkhorn iterations [51]. For the self-attention, we use fully-connected graph for the first five layers before changing the number of $k$ nearest neighbors in deeper layers. More specifically, in the last four layers, we set $k_6^{self} = k_7^{self} = 128$, $k_8^{self} = k_9^{self} = 64$. In terms of cross-attention, we keep all fully connected layers, where $k_{1-9}^{cross} = 256$. The model is implemented in the Pytorch framework and NVIDIA GTX 3090.

### 4.3. Training

Training is performed on the KITTI odometry dataset [56]. The ground-truth results obtained by GPS are also included. To reduce the noise in each pair of scans, we apply ICP to refine the alignment. If registration fails or the number of matching voxels is less than 1k, we filtered that pair out of the preprocessed dataset. In addition, we only select pairs with a minimum distance of 10 m away from each other for training and testing. By doing this, we yield more samples, which are 17,920 for training, 4070 for validation, and 1200 for testing. The model is optimized using the Adam method [57] with a batch size of 64. The learning rate is set at 0.001 during the first 15 epochs and decreases to 0.0001 until convergence.

## 5. Experiments

We validate the DFGAT on the outdoor dataset. Additionally, we compare our evaluations to handcrafted descriptors. Our approach can better distinguish the features in registration on the KITTI odometry dataset.

### 5.1. Evaluation Metrics

To measure the matching performance, we compute the matching Precision, Accuracy, Recall and F1-score from the ground-truth correspondences. According to registration ability, we use the indicators proposed in [58,59]: relative rotation error (RRE), relative translation error (RTE), and success rate.

*RTE* and *RRE* are defined as:

$$RTE = |\hat{T} - T^*|$$
$$RRE = arccos\frac{Tr(\hat{R}^T R^*) - 1}{2}$$

(9)

where $T^*$ and $R^*$ denote the ground-truth translation and rotation, respectively. Meanwhile, $\hat{T}$ is the estimated translation and $\hat{R}$ is the estimated rotation matrix after registration.

*5.2. Performance*

The matching is evaluated as follows. For fair comparisons, we refer to the results of the group of methods including USIP detector [42], MDGAT descriptor and SuperGlue matcher. Specifically, we re-check SuperGlue matcher with several threshold assignments algorithm and threshold-free. After that, we run MDGAT-matcher instead of the combination between MDGAT descriptor and SuperGlue matcher. As show in Table 1 and Figure 3, the DFGAT achieves the best performance in matching ability with about 80% for Precision, 89% for Accuracy, 76% for Recall and 0.783 for the F1-score. Although the USIP can detect keypoints by using an unsupervised learning network, it struggles to predict keypoints with a limited number of data and density-invariant requirements. The further reason is that the USIP method is not integrated with the descriptions framework which limits the full exploitation of the geometric characteristics of the point cloud.

**Table 1.** Matching performance.

| Matcher | Precision | Accuracy | Recall | F1-Score |
|---|---|---|---|---|
| MDGAT Desc. + SuperGlue(0.1) | 54.3 | 76.7 | 57.7 | 0.559 |
| MDGAT Desc. + SuperGlue(0.2) | 57.6 | 75.8 | 49.2 | 0.531 |
| MDGAT Desc. + SuperGlue(0.3) | 62.2 | 72.7 | 39.4 | 0.482 |
| MDGAT Desc. + SuperGlue(0) | 57.7 | 72.2 | 35.4 | 0.439 |
| MDGAT (Desc. + matcher) | 70.3 | 86.7 | 69.4 | 0.698 |
| **DFGAT** | **80.4** | **90.2** | **76.1** | **0.782** |

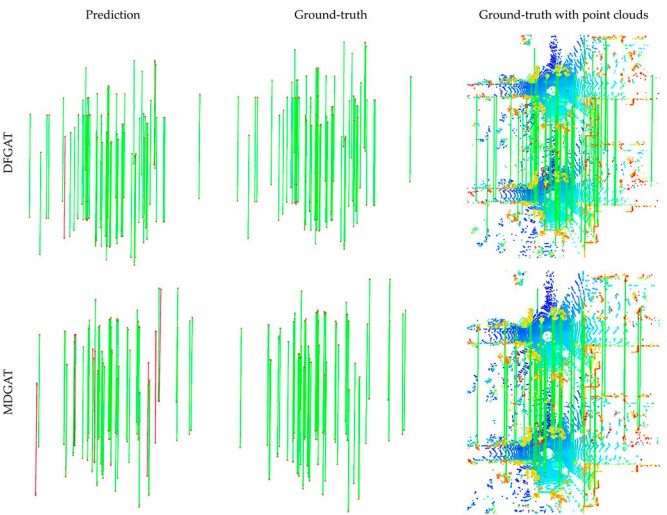

**Figure 3.** Visualization of matching results on KITTI. The first column are keypoints detected and matched using our method and MDGAT. The second column are ground-truth results without raw point clouds while the third column represents ground-truth results including raw point clouds. Our method performs better in terms of keypoints matching with more correct pairs (green) and fewer wrong pairs (red).

We further evaluate the registration ability of the DFGAT as shown in Table 2. We compare the metric scores achieved against several strong baselines: the hand-crafted

methods RANSAC [23], FPFH [24], SHOT [30], and USIP. For the learning-based method, theMDGAT + SuperGlue [4] is selected. As shown in Table 2, our network achieves the lowest registration failure rate of 0.117% among all the baselines and has a comparable inlier ratio to that of the MDGAT-matcher. The reason is that our descriptor can create better learnable features, and after the Sinkhorn algorithm assignment, the scores matrices have many more positive pairs. For fair comparison with MDGAT, our approach uses the number of desired keypoints 256 and has large improvements in RTE, RRE, and the success ratio. In addition, as shown in Table 3, our model can achieve the lowest score (0.24°) on the RRE scale and the highest percentage success (99.88%), while the D3Feat still remains the best in RTE with 6.90 cm. It is noteworthy that our approach uses only 256 keypoints for each point cloud, whereas in other methods, they originally used no less than 1024 keypoints. In terms of training time, our model takes only 40 epochs in comparison with 1000 epochs for the MDGAT-matcher because we use better strategy in the keypoints selection and description.

**Table 2.** Registration performance.

| Matcher | Failure Rate | Inlier Ratio |
|---|---|---|
| FPFH + RANSAC | 8.37 | 18.77 |
| SHOT + RANSAC | 5.40 | 18.21 |
| USIP + RANSAC | 1.41 | 32.20 |
| MDGAT Desc. + SuperGlue | 0.58 | 36.19 |
| MDGAT (Desc. + matcher) | 1.33 | **37.54** |
| **DFGAT** | **0.117** | 34.93 |

**Table 3.** Registration results on KITTI.

| Method | RTE (cm) | Standard Deviation (cm) | RRE (°) | Standard Deviation (°) | Success (%) |
|---|---|---|---|---|---|
| 3DFeat-Net | 25.9 | 26.2 | 0.57 | 0.46 | 95.97 |
| FCGF | 9.52 | 1.30 | 0.30 | 0.28 | 96.57 |
| D3Feat | **6.90** | **0.30** | **0.24** | **0.06** | 99.81 |
| SpinNet | 9.88 | 0.50 | 0.47 | 0.09 | 99.10 |
| MDGAT-matcher | 16.00 | 64.9 | 0.608 | 3.56 | 98.67 |
| **DFGAT** | 9.70 | 21.0 | **0.24** | 1.22 | **99.88** |

### 5.3. Ablation Study

We perform ablation studies on three key components of our framework: the descriptor encoder, the GAT, and the loss function.

#### 5.3.1. Ablation of Descriptor Encoder

In this experiment, we compare four different combinations of descriptor encoders. The first option is the PointNet [60] encoder, working on separate points sampled by the nearest neighbor search. The second design takes the output of descriptor decoder and performs max pooling to extract global features. We then use an MLP to embed this information into a global descriptor as the input of GAT. We refer to this descriptor as Global Desc. The third design is the one proposed by Shi et al. [4] named MDGAT. This design is inspired by the USIP detector [42] and FPFH descriptor [24] to enrich the feature representation before being fed into the GAT.

We train different networks with the above encoders. The results in Table 4 demonstrate that our descriptor with most generalized structure can achieve the best performance. The result is that the density-invariant feature descriptor inspired by D3Feat allows the network to cope with the inherent density of point cloud variations. Furthermore, the dynamic GAT with two types of attention enhances the performance of a powerful and flexible matching mechanism. For Global Desc., we can see that concatenating global information too early in the descriptor encoding process does not improve the matching efficiency but

even reduces the discriminant ability. The same effect applies to PointNet, which extracts the overall interface using maximum aggregation and loses local information.

**Table 4.** Ablation of descriptor encoder.

| Descriptor Encoder | Precision | Accuracy | Recall | F1-Score |
|:---:|:---:|:---:|:---:|:---:|
| PointNet | 59.4 | 80.3 | 62.3 | 0.608 |
| Global Desc. | 79.1 | 89.7 | 75.2 | 0.771 |
| MDGAT | 70.3 | 86.7 | 69.4 | 0.698 |
| **DFGAT** | **80.4** | **90.2** | **76.1** | **0.782** |

5.3.2. Ablation of GAT

In this study, we evaluate several variations of a dynamic GAT with different decay strategies as the network goes deeper into the last layers. We introduce the DFGAT variant 1 using $k_{1-7}^{self} = 128$ for the first 7 layers and $k_{8,9}^{self} = 64$ for the last 2 layers in self-attention. For cross-attention, we maintain full connection for all 9 layers with $k_{1-9}^{cross} = 256$. We refer to the other design using the first 5 fully connected layers, while 6,7 layers have $k_{6,7}^{self} = k_{6,7}^{cross} = 128$, and the last 2 layers $k_{8,9}^{self} = k_{8,9}^{cross} = 64$, as for DFGAT variant 2.

As shown in Table 5, compared with two above variant versions, our gradual decay strategy performs best. The reason could be that first examining the large observations and then gradually focusing on specific interests becomes more natural. Specifically, self-attention starts from the full connected layer and $k$ gradually decreases in the last layers. In addition, the cross-attention will always maintain the full connected layer to ensure the generalization information between connections.

**Table 5.** Ablation of GAT.

| Matcher | Precision | Accuracy | Recall | F1-Score |
|:---:|:---:|:---:|:---:|:---:|
| DFGAT variant 1 | 77.9 | 89.6 | **76.3** | 0.771 |
| DFGAT variant 2 | 78.2 | 89.1 | 73.6 | 0.758 |
| **DFGAT** | **80.4** | **90.2** | 76.1 | **0.782** |

5.3.3. Ablation of Loss Function

In this ablation study, we evaluate the gap loss against the SuperGlue loss [53] and the triplet loss [32] using the hardest negative match. We retrain our matcher on three loss functions and show the results in Table 6. In comparison with the SuperGlue loss, the gap loss is more suitable for the matching highly density point clouds problem. In addition, gap loss outperforms triplet loss due to its ability to use all the negative matches more efficiently to improve generalization.

**Table 6.** Ablation of loss function.

| Loss | Precision | Accuracy | Recall | F1-Score |
|:---:|:---:|:---:|:---:|:---:|
| SuperGlue loss | 54.6 | 75.7 | 47.6 | 0.509 |
| Triplet loss | 56.3 | 77.8 | 51.6 | 0.538 |
| **Gap loss** | **80.4** | **90.2** | **76.1** | **0.782** |

## 6. Discussion

Robotics and autonomous driving both face a basic challenge with point cloud registration. It is used in medical imaging, panorama stitching, object localization and recognition, motion estimates, 3-D reconstruction, and object recognition. Common methods for registration with unknown or uncertain correspondences depend on the existence of an initial hypothesis for the unknown transformation. These methods could stop working at any time and return estimates that are arbitrarily far from the transformation of the ground

truth. In our proposed study, the keypoint detector, feature descriptor, and the dynamic graph attention network are the three components of our proposed DFGAT. The detector uses deep learning to achieve repeatability, specificity, and computing efficiency in its capacity to identify key locations. The learned descriptor can preserve intricate local geometric patterns and is rotation invariant. The graph architecture focuses on useful information with self and cross-attention. To demonstrate the efficiency and uniqueness of our network, experiments using the KITTI odometry are carried out. Our methodology surpasses previous registration methods with a success ratio of almost 99.88 % and achieves the best results in evaluation matching metrics when compared with existing methods.

## 7. Conclusions

In this study, we present a GAT-based framework for point cloud registration. Our proposed DFGAT includes three components: a keypoint detector, a feature descriptor, and a dynamic graph attention network. The detector is deep learning-based so that it can detect key points with repeatability, particularity, and computational efficiency. The learned descriptor is rotation invariant, descriptive, and able to preserve complex local geometric patterns. The graph architecture with self- and cross-attention focuses on the useful information. Experiments with the KITTI odometry are conducted to show the effectiveness and distinctiveness of our network in comparison with the state-of-the-art approaches. Among existing matching methods, including both learned and traditional ones, our approach achieves the best results in evaluation matching metrics and outperforms other registration approaches with a success ratio of 99.88% and the lowest failure rate of 0.117%. The results suggest that our approach outperforms similar existing methods and achieves substantial improvements. An interesting avenue for future work is to integrate detector, descriptor, and graph networks into an end-to-end architecture.

**Author Contributions:** Conceptualization, Q.-V.L.-D.; data curation, Q.-V.L.-D. and H.J.; formal analysis, Q.-V.L.-D., S.H.N. and H.J.; investigation, Q.-V.L.-D.; methodology, Q.-V.L.-D. and S.H.N.; software, Q.-V.L.-D.; supervision, Q.-V.L.-D. and S.H.N.; visualization, Q.-V.L.-D. and S.H.N.; writing—original draft, Q.-V.L.-D.; writing—review & editing, Q.-V.L.-D., S.H.N. and H.J. All authors have read and agreed to the published version of the manuscript.

**Funding:** This work was funded by the Korean government (MSIT) (No. 2020-0-00440).

**Institutional Review Board Statement:** Not applicable.

**Informed Consent Statement:** Not applicable.

**Data Availability Statement:** Not applicable.

**Acknowledgments:** This work was supported by an Institute for Information Communications Technology Promotion (IITP) grant funded by the Korean government (MSIT) (No. 2020-0-00440, Development of Artificial Intelligence Technology that continuously improves itself as the situation changes in the real world).

**Conflicts of Interest:** The authors declare no conflict of interest.

## Abbreviations

| | |
|---|---|
| GAT | Graph attention network |
| DFGAT | Dense features graph attention network |
| SLAM | Simultaneous localization and mapping |
| WSNs | Wireless sensor networks |
| LiDAR | Light detection and ranging |
| ICP | Iterative closest point |
| RANSAC | Random sample consensus |
| LRF | Local reference frame |
| D3Feat | Dense detection and description of 3d local |

| KPConv | Kernel point convolution |
| LBSS | Laplace-Beltrami Scale-space |
| USIP | Unsupervised stable interest point |
| PPF | Point-pair features |
| FPFH | Fast point feature histograms |
| ISS | Intrinsic shape signatures |
| SHOT | Signature of histograms of orientations |
| FCGF | Fully convolutional geometric features approach |
| MLP | Multi-layer perceptron |

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
