# Peer review of "Learning Dense Features for Point Cloud Registration Using a Graph Attention Network"

_applsci, doi:10.3390/app12147023_

Round 1

Reviewer 1 Report

It is a good practice to combine several authorisms together to solve the problem of point cloud registration. There are some comments to the authors:

1.       The claim for computation capacity limitation in the abstract should be changed since the parallel computing with GPU has compensated this disadvantage for many algorithms for example ICP.

2.       Line 39 page 1. Structure sensor, this is not an example of the handheld depth sensors, instead, it should be a catalog of the sensors. Besides, give the appropriate reference for each product.

3.        KITT. It needs a reference to give KITT a credit if you use it.

4.       Line 66, page 1. ICP and its variants (please give some reference for the variants)

5.       Fig1. The abbreviations should be given the original term for example V2N, V2V …. It is better for readers who don’t familiarize with the WSN and communication.

6.       The descriptions to the symbols used in equation 1 are not accurate. For example NiQ is a set of neighbor.

7.       For dynamic GAT, please give the appropriate reference to show the efficiency.

8.       Please make sure the losing function 8 is in accurate form.

9.       Page 8, line 306. I was confused to use ICP to reduce the noise. Please give more details.

10.   Indicate the layers of NN you used with dynamic GAT.

11.   Pictures are blurry. Please update the pictures.

Author Response

Dear Editor and Reviewers,

Thanks to the Editor for guiding review process and anonymous reviewers for their valuable comments that help improve the quality of the draft. Please find the attached file.

Reviewer 2 Report

In the study, a point cloud registration method based on deep learning is presented. The work is well organized and clearly explained. I offer some suggestions for the enhancement of the work.

1) The Fast Point Feature Histogram is a descriptor than a detector. It should be examined under descriptors in the literature review.

2) Authors should explain on what basis they set the parameters. I suggest you perform an ablation study examining the change in these parameters.

3) Authors should introduce the dataset before the parameters.

4) It should be discussed whether the results of the proposed method make a significant improvement compared to the similar ones in the literature. The Discussion section should be expanded considering the accuracy parameters.

Author Response

(The authors gave the same response as above.)

Round 2

Reviewer 1 Report

The authors address my concerns. 

Reviewer 2 Report

Necessary corrections were made in the paper. It is now available for publication.